# The Cultivation Techniques and Quality Characteristics of a New Germplasm of *Vitis adenoclada* Hand.-Mazz Grape

**Dai-Dong Wu [1], Guo Cheng [1,\*], Hong-Yan Li [1], Si-Hong Zhou [1], Ning Yao [2], Jin Zhang [1] and Lin-Jun Xie [1]**

[1] Grape and Wine Research Institute, Guangxi Academy of Agricultural Sciences, Nanning 530007, China; Wu0089@126.com (D.-D.W.); xiaoyan821025@163.com (H.-Y.L.); bear824@126.com (S.-H.Z.); zhangj@gxaas.net (J.Z.); xielinjun@gxaas.net (L.-J.X.)

[2] Guangxi Luocheng County *Vitis heyneana* Roem. & Schult, Grape Experimental Station, Hechi 546400, China; yaoning001@126.com

\* Correspondence: berry713@gxaas.net; Tel.: +86-1376-811-9416

**Abstract:** *Vitis adenoclada* Hand.-Mazz is a special wild grape resource that is often confused with *Vitis heyneana* Roem. & Schult in research or production practice, and there are few comprehensive studies on this species in recent years. "Gui Heizhenzhu No. 3" (GH3), as a new germplasm of *V. adenoclada* found in Guangxi, China, has many advantages, such as good quality and strong adaptability. In this paper, an attempt was made to introduce the breeding process of GH3, including a brief description of its botanical characteristics and its cultivation and management techniques in karst rocky desertification mountains. Meanwhile, its quality-related parameters were evaluated by widely targeted metabolomic analysis. This study indicated that GH3 had the typical botanical characteristics of *V. adenoclada*, but with larger fruit and a higher sugar content compared to wild or other *V. adenoclada* grape varieties. Metabolomic study of the target variety showed that glucose and citric acid were the main sugar and acid components in fully ripened berries. Moreover, cyanidin-3-*O*-glucoside presented as the characteristic anthocyanin. In addition, B-ring dihydroxylation was more active than trihydroxylation in the GH3 berry. Several of its botanical and quality characteristics highlight the unique genetic background of this variety. Thus, it has an important guiding significance and a scientific theoretical basis for identifying, exploiting, and utilizing East Asian wild grape resources.

**Keywords:** *V. adenoclada* Hand.-Mazz; GH3; breeding; cultivation techniques; quality characteristics

## 1. Introduction

China is one of the countries of origin of the grape genus *Vitis* and is one of the most abundant germplasm resources in the world. The genetic basis of cultivated grapes is narrow, and the wild grape relative species contains a large number of excellent genes urgently needed by cultivated grapes, making it an important gene pool to broaden the genetic basis of cultivated grapes [1]. East Asian wild grape species contain many varieties, which present strong stress resistance and good agronomic traits. Cross-breeding with cultivated grape varieties using wild resources with close relatives can obtain excellent table grape varieties, resistant resources, or wine grape varieties. *V. adenoclada* is a unique wild grape species in China, classified as an East Asian species and widely distributed in Hunan, Fujian, Guangxi, and other provinces in the south of the Yangtze River [2]. For a long time, *V. adenoclada* was mistaken as *V. heyneana* Roem. & Schult, which created significant inconvenience to the field of research on the utilization of wild grape resources, especially in terms of hybridization [3]. Therefore, there have been few reports about *V. adenoclada* in recent years.

Luocheng county, Hechi city, Guangxi province, China, is located in eastern Asia (24°47′24″ N, 108°54′00″ E), which has a subtropical climate and mountains. It is a typical topographical and geomorphic region in a karst mountain area and is the main origin of the East Asian *Vitis*. In 2011, the natural hybrid seeds of five lines of the pistillate flower of *V. adenoclada*, such as Shuiyuan No. 1, Shuiyuan No. 11, Jiulong No. 1, Cotton No. 5, and Lile No. 6, were seeded together in the Shuiyuan wild grape base in Luocheng county, and 139 seedlings were planted in an open field. No chemical pesticides were used throughout the growing season other than Bordeaux solution to prevent grape downy mildew. The strain numbered CHN-GX-GH-14-03 is the new germplasm of *V. adenoclada* used in this study (hereinafter referred to as "Gui Heizhenzhu No. 3" or "GH3" for short).

GH3, selected by our research team, is available for table use and processing, belonging to the woody vine plant. The GH3 grape has a unique honey-smelling taste for its table use, and it can adapt to climatic conditions, such as the high temperature, high humidity, abundant rain, and little sunshine in southern China as well as other *V. adenoclada* grapes [2]. It presents excellent resistance to a wet and hot climate under artificial cultivation, as well as strong tolerance to drought and poor soil, especially in rocky desertification karst mountain areas [4]. The strength and vitality of the root system can take advantage of the deep soil moisture and fertility, and the climbing vines can cover bare rocky hills and green barren hill slopes, can purify the air, reduce soil erosion, and promote the diversity of the ecological environment. These features are similar to previous relevant research in karst mountain regions [5].

*V. adenoclada* shows excellent performance in disease, water, heat, and drought resistance, as well as high yielding and photosynthetic ability [2,3,6,7]. Most previous studies focused on the identification, evaluation, genetic diversity, and agronomic characteristics of *V. adenoclada* [2,8]. Unfortunately, there is little comprehensive research of its breeding, cultivation techniques, and quality characteristics. In the present study, we describe the botanical and biological characteristics of the new germplasm GH3 (*V. adenoclada*) grape. The key techniques of cultivation in rocky desertification karst mountain areas in China include land selection, training system, pruning, fertilizer and water management, and disease and pest control. Meanwhile, the quality-related parameters, such as soluble sugars, organic acids, and phenols, of the variety, were also analyzed.

## 2. Materials and Methods

### 2.1. The Experimental Site and Plant Material

The experimental field was located in Luocheng county (belonging to Hechi city), Guangxi province, China. The vines were planted in 2012 and grown on their own roots in an east–south row orientation at an altitude of 280 m. All of the vines were trained to a high hanging trellis system, spaced at 6.0 × 5.0 m (irregular distribution because of the rocky mountain; Figure 1). The crop load was normalized to approximately 300 bunches per plant.

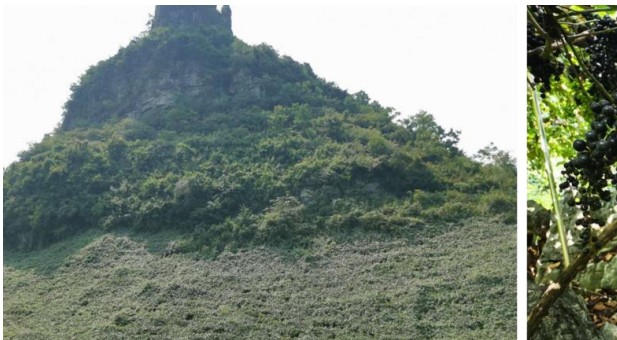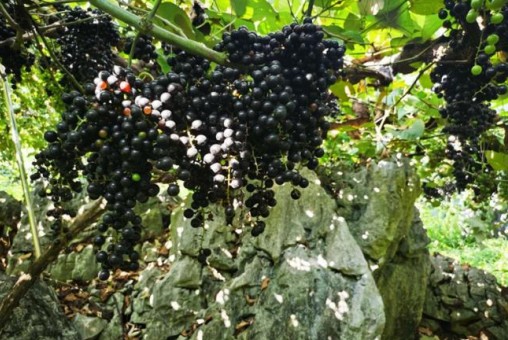

**Figure 1.** Gui Heizhenzhu No. 3 (GH3), growing on rocky mountains (**left**) and in the crevices of the rocks (**right**). Notice the large size of the rocks around the fruiting vines at the bottom right. Photos by Dai-Dong Wu.

## 2.2. Description of the Variety's Characteristics

The botanical characteristics were studied mainly through field observation, photographing, measurement, and recording. The botany terms and definitions recorded in the book *The Science of Grapevines: Anatomy and Physiology* [9] were adopted. The phenological descriptions were attributed in accordance with Eichhorn and Lorenz reproduced from Coombe [10].

## 2.3. A Brief Technique for the Cultivation and Management of GH3 in Karst Rocky Desertification Mountains

### 2.3.1. Selection of Land, Preparation of Land, Varieties, and Planting

GH3 is highly adaptable to various ecological conditions, such as the soil and climate south of the Yangtze River. As long as there is a certain amount of soil, GH3 can be planted in hilly, mountainous, rocky areas and under other site conditions. Three hundred grapevines were planted per hectare, and each plant had at least $0.5 \text{ m}^3$ of soil for cultivation. Vegetative cup green leaf seedlings should be planted from April to June, while naked root seedlings should be planted before or after budbreak from February to April. Before transplanting, 0.5 kg of phosphate fertilizer was mixed with 2–5 kg of high-quality organic fertilizer. After transplanting, irrigation was required, and then an appropriate amount of water-retaining agent (10 g per plant) was mixed into the soil of the transplanting holes to prevent drought and to moisturize and promote growth.

### 2.3.2. Training System and Pruning

High-hanging trellis system: In hilly and rocky depressions with large slopes, it is advisable to use a large, well-shaped high hanging frame for cultivation. The horizontal bearing line is a single line with a distance of 2.0–3.0 m. The main tendrils are tied along the horizontal line, and fruiting branches are free to hang down and grow. Flat canopies can be built on gentle hills.

Ecological trellis system: Stone mountains, rocky desertification mountains, and low forest land can directly pull the branches and vines to nearby surfaces, so an ecological frame requires less investment but high efficiency.

Training: A high hanging frame and flat trellis are similar; all first-class vines grow vertically, while all second-class vines grow horizontally. All secondary main vines are grown from the vertical first-class vines, while the bearing branches are directly cultivated on the first- and second-class vines every 30–40 cm on each side.

Winter pruning: Usually, in the middle of January to early March, $6\text{–}12 \text{ buds/m}^2$ and $6\text{–}10 \text{ shoots/m}^2$ are pruned. Pinching is an important measure to protect flowers and fruits 5–7 days before flowering. Generally, 1–3 clusters are left for each bearing shoot, and the spacing of new shoots is approximately 25 cm to prevent too much crossing and crowding. After harvesting, overdense and insect-damaged and diseased branches are cut off to restore tree vigor as soon as possible.

### 2.3.3. Fertilizer Management

After being put into production, grapevines are usually fertilized twice a year. In the first instance, 0.67–2.00 kg/ha of a nitrogen–phosphorus–potassium mixed fertilizer was applied during the expansion period of young grape fruits, within $2\text{–}3 \text{ m}^2$ of the tree plate after rain. In the second instance, 33.33–66.67 kg of organic fertilizer + 0.13–0.27 kg of phosphate fertilizer, as well as 0.03–0.07 kg of nitrogen fertilizer + 0.03–0.07 kg of potash fertilizer, were applied per hectare after harvest. It is recommended that the use of medium and trace elements be supplemented conditionally according to the results of a soil test year after year to rotate the deep tillage position.

### 2.3.4. Pest Control

GH3 showed strong resistance to diseases and insects, except for downy mildew. Thus, the corresponding measures in production mainly focus on the prevention of downy mildew. The specific

preventive measures of each growth period were as follows: From dormancy to the beginning of the budbreak period, the whole vineyard was extensively cleared, and when brown wool was visible in the buds (E-L3), a 45% sulfur mixture 300 times of sulfur mixture was sprayed on the trunk, branches, and ground for reducing the base of various diseases and insect pests. Downy mildew easily occurred in the bud and leaf spreading stage, before flowering, after flowering, and in the young fruit swelling stage. Fungicides with preventive and therapeutic effects can be applied alternately. After expansion to the early stage of maturity, a 1:1:200 Bordeaux mixture, thyme, or mycezole was sprayed and rotated for 2–3 times. GH3 has strong growth potential and a strong self-compensation ability, and pest damage was relatively light across the whole growing season.

### 2.4. Berry Sampling and Physical Chemical Index Analysis

Grape berries in three biological replicates were collected in five E-L stages [10] in 2020: Berries still hard and green (E-L 33); berries beginning to soften: Brix start increasing (E-L 34); the onset of veraison (E-L 35); berries not quite ripe (E-L 37); the harvest stage (E-L 38). For each biological replicate, 150 berries were randomly separated from at least 50 clusters within 6 vines. The sampling time was fixed at 9:00–10:00 a.m., and three biological replicates were collected via the same method at each sampling date. After being transported to the laboratory, a subsample of 100 berries from each biological replicate was subjected to physiological measurements, including berry fresh weight, total soluble solids (TSS) content, pH, and titratable acidity (TA). The remaining fruit berries (50 berries from each biological replicate) were stored at −80 °C after quick-freezing in liquid nitrogen, but only the fifth sampling point was used for the subsequent metabolomic analysis. Moreover, the color parameters were analyzed using a colorimeter after weighing (Konika Minolta CR-10, Japan). At the harvest stage, 300 berry samples (100 per replicate) were crushed in a hand press through two layers of cheesecloth after weighing. The berry skins were obtained by carefully removing the seeds and mesocarp. The berry skins were then rinsed in deionized water and weighed after blotting the excess water. TSS was measured using a PAL-1 Digital Handheld "Pocket" Refractometer (Atago, Tokyo, Japan). TA was measured by titration with NaOH to the endpoint of pH 8.2 and expressed as tartaric acid equivalents.

In 2018 and 2019, we also carried out some physical and chemical parameters of GH3 berries at harvest stages, mainly including berry fresh weight, TSS, pH, and TA. To better understand the characteristics of the new grapevine genotype GH3, we provide this part of the data in Supplement Table S1.

### 2.5. Widely Targeted Metabolomic Analysis

#### 2.5.1. Sample Preparation and Extraction

The freeze-dried berries at harvest were crushed using a mixer mill (MM 400, Retsch) with a zirconia bead for 1.5 min at 30 Hz. Then, 100 mg powder was weighed and extracted overnight at 4 °C with 1.2 mL of 70% aqueous methanol. Following centrifugation at 12,000 rpm for 10 min, the extracts were filtrated (SCAA-104, 0.22 μm pore size; ANPEL, Shanghai, China; http://www.anpel.com.cn/) before UPLC–MS/MS analysis.

#### 2.5.2. UPLC Conditions

The sample extracts were analyzed using a UPLC–ESI–MS/MS system (UPLC, Shim-pack UFLC SHIMADZU CBM30A system, www.shimadzu.com.cn/; MS, Applied Biosystems 4500 Q TRAP, www.appliedbiosystems.com.cn/). The analytical conditions were as follows: UPLC column, AgilentSB-C18 (1.8 μm, 2.1 × 100 mm); the mobile phase consisted of solvent A (pure water with 0.1% formic acid) and solvent B (acetonitrile). Sample measurements were performed with a gradient program that employed the starting conditions of 95% A and 5% B. Within 9 min, a linear gradient to 5% A and 95% B was programmed, and a composition of 5% A and 95% B was kept for 1 min. Subsequently, a composition of 95% A and 5.0% B was adjusted within 1.10 min and kept for 2.9 min.

The column oven was set to 40 °C, and the injection volume was 4 μL. The effluent was alternatively connected to an ESI–triple quadrupole linear ion trap (QTRAP)–MS.

### 2.5.3. ESI–QTRAP–MS/MS

LIT and triple quadrupole (QQQ) scans were acquired on a QTRAP mass spectrometer (API 4500 Q TRAP UPLC/MS/MS System) equipped with an ESI turbo ion spray interface, operating in positive and negative ion modes and controlled by Analyst 1.6.3 software (AB Sciex). The ESI source operation parameters were as follows: Ion source and turbo spray; source temperature of 550 °C; ion spray voltage (IS) of 5500 V (positive ion mode)/–4500 V (negative ion mode); ion source gas I (GSI), gas II (GSII), and curtain gas (CUR) set at 50, 60, and 30.0 psi, respectively; high collision gas (CAD). Instrument tuning and mass calibration were performed with 10 and 100 μmol/L of polypropylene glycol solutions in the QQQ and LIT modes, respectively. The QQQ scans were acquired as MRM experiments with the collision gas (nitrogen) set to 5 psi. The DP and CE for the individual MRM transitions were conducted with further DP and CE optimization. A specific set of MRM transitions were monitored for each period according to the metabolites eluted within this period.

### 2.6. Statistical Analysis

Significant differences were determined when $p < 0.05$ according to Duncan multiple comparison. Statistical analysis was performed by SPSS (SPSS Inc., Chicago, IL, USA) for Windows, version 20.0.

## 3. Results

### 3.1. The Variety's Characteristics

#### 3.1.1. Botanical Characteristics

GH3 is a female flower plant with strong growth potential. New shoots were densely covered with gray arachnoid tomentum, as well as densely purple–brown glandular hairs at the base, gradually thinning upward. The upper branchlets had no glandular hairs. The leaf blade was oval, ovate–oblong, or ovatequinquangular and 16.61 × 12.82 cm in size, with an acuminate apex, a cordate to subcordate base, and a 21–28-toothed margin on each side (Figure 2 and Table 1). The upper surface of the leaves was sparsely covered with arachnoid tomentum when young, then glabrate. The lower surface of the leaf blade was densely covered with gray tomentum. The average length of the petiole was 8.28 cm. The tendrils were branched and densely tomentose and grow intermittently. The average number of inflorescences per branch was approximately 3.3, while the fruit branch rate was approximately 91% (field survey, but data not shown). The average cluster length and width were 13.7 cm and 8.04 cm, respectively (Table 1). The average cluster weight was 132.3 g, and the maximum cluster weight was up to 323.6 g (Table 1). The average longitudinal diameter and transverse diameter of the berries were 1.59 cm and 1.49 cm, respectively (Table 1). The berries were purple–black at maturity, globose, and 1.07 in the shape index (Table 1). The average berry weight was 2.33 g, with 3–5 seeds per berry and an average seed number of 4.16 (Table 1). The berry skin weight was 0.39 g, and the skin to berry ratio was 16.89 (Table 1). The seeds were obovoid, the apex was rounded, the chalazal knot was rounded, and the ventral holes furrowed upward a quarter from base.

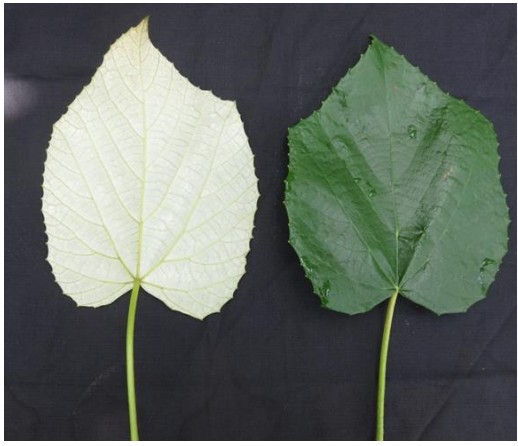 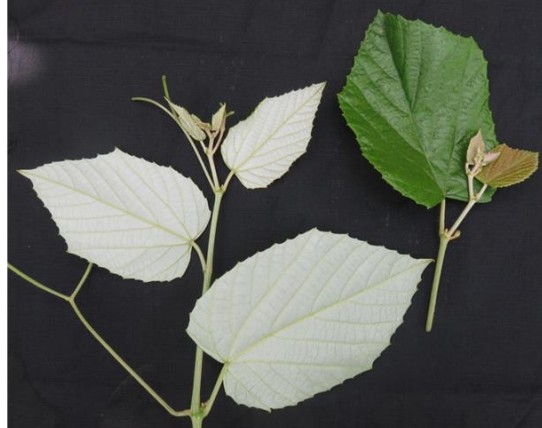

**Figure 2.** Front and back of a leaf of GH3 (left) and leaves grown on new shoots (right). Photos by Dai-Dong Wu.

**Table 1.** The leaf, cluster, and berry indicators of Gui Heizhenzhu No. 3 (GH3).

| Organ | Indicator | Value |
|---|---|---|
| Leaf | Leaf length | 16.61 ± 1.43 |
| | Leaf width | 12.82 ± 0.99 |
| | Petiole length | 8.28 ± 1.26 |
| Cluster | Cluster length | 13.70 ± 1.60 |
| | Cluster width | 8.04 ± 1.63 |
| | Cluster weight | 132.30 ± 19.48 |
| Berry | Longitudinal diameter | 1.59 ± 0.03 |
| | Transverse diameter | 1.49 ± 0.02 |
| | Shape index | 1.07 ± 0.01 |
| | Berry weight | 2.33 ± 0.06 |
| | Number of seeds | 4.16 ± 0.15 |
| | Skin weight | 0.39 ± 0.02 |
| | Skin to berry ratio | 16.89 ± 0.73 |

### 3.1.2. Phenological Period

Budbreak takes place in early April, bloom in late May (Figure 3), veraison in mid-August, and fruits were fully ripe in mid-to-late September. It took approximately 150–160 days from budbreak to full fruit ripening, and it was a late-ripening variety. Figure 4 shows the growth stages of the GH3 grapevine.

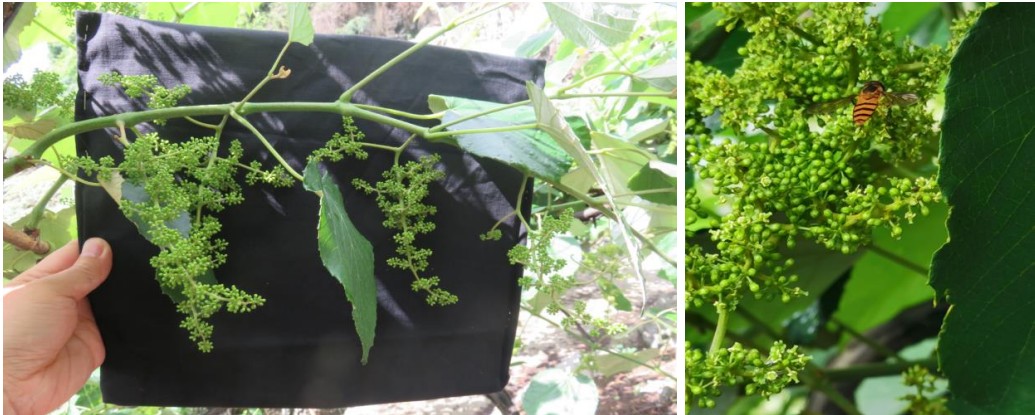

**Figure 3.** GH3 inflorescence at the beginning of bloom (**left**) and in full bloom (**right**). Photos by Dai-Dong Wu.

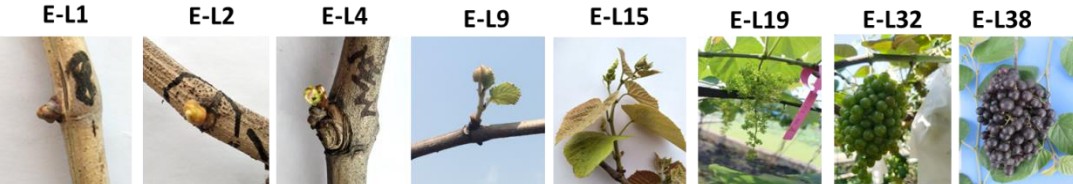

**Figure 4.** GH3 grapevine growth stages, according to Eichhorn and Lorenz. Photos by Dai-Dong Wu.

### 3.1.3. Stress Resistance

The leaves, shoots, and fruits were highly resistant to anthracnose and were almost immune under outdoor cultivation in southern China with high temperatures, high humidity, and rainy weather. They also had high resistance to white rot, grey mold, brown leaf spot, rust, and powdery mildew, as well as moderate resistance to downy mildew. Downy mildew occurs easily in inflorescence, before and after flowering to the swelling stage of young fruit, but rarely in leaves and branches. This variety also had wide adaptability to soil. In the south of China, with the high temperature, high humidity, and rainy areas of plant growth, GH3 grapevines presented strong bearing capacity. Therefore, GH3 can be used as a new germplasm for fresh food and processing.

### 3.2. Grape Berry Development

Samples that corresponded to the five stages of the E-L system were measured for berry fresh weight, TSS, pH, TA, and some color parameters (Table 2). As grapes ripen, their berry fresh weight, TSS content, and pH gradually increased, and their titratable acidity content decreased. It is worth noting that in this study, the TSS content of the berries increased, but the titratable acidity content decreased sharply in the veraison stage (E-L35), indicating that the fruit of GH3 was well ripened. A gradual decrease in the brightness value L* indicates a darker skin color as the development progressed. Meanwhile, an increase in the green-red hue (a*) means that the skin color changed from green to red. Lastly, a decrease in the yellow-blue hue indicates a gradual lightening of the yellow hue.

**Table 2.** Physical chemical and color parameters of GH3 berries from the five developmental stages.

| Parameters | Developmental Stages | | | | |
|---|---|---|---|---|---|
| | E-L 32 | E-L 33 | E-L 35 | E-L 37 | E-L 38 |
| Berry fresh weight (g) | 1.33 ± 0.03 d | 1.41 ± 0.03 d | 1.60 ± 0.05 c | 1.79 ± 0.31 b | 2.33 ± 0.06 a |
| Total soluble solids (°Brix) | 4.67 ± 0.15 e | 7.37 ± 0.15 d | 13.80 ± 0.10 c | 16.80 ± 0.10 b | 21.10 ± 0.10 a |
| pH | 2.11 ± 0.07 d | 2.14 ± 0.03 d | 2.72 ± 0.03 c | 2.89 ± 0.06 b | 3.16 ± 0.06 a |
| Titratable acidity (g/L) | 53.40 ± 0.53 a | 51.19 ± 0.75 b | 17.71 ± 0.84 c | 12.43 ± 0.24 d | 5.06 ± 0.07 e |
| L* | 42.36 ± 0.23 a | 41.26 ± 0.67 a | 35.87 ± 0.95 b | 26.66 ± 0.16 c | 23.75 ± 1.12 d |
| a* | −2.83 ± 0.15 e | −2.67 ± 0.15 d | −0.43 ± 0.67 c | 1.79 ± 0.19 b | 1.91 ± 0.18 a |
| b* | 19.81 ± 0.45 a | 16.90 ± 0.30 b | 12.86 ± 1.02 c | 2.81 ± 0.14 d | 2.82 ± 0.13 d |

Note: Values within a row followed by different lowercases differ significantly among the five developmental stages (Duncan, $p < 0.05$).

### 3.3. Metabolism and Component Characteristics of Sugars, Organic Acids, and Phenolic Compounds

The sugars, organic acids, and phenolic compounds were analyzed at harvest to explore the quality characteristics of the new germplasm *V. adenoclada* GH3. The soluble sugar and organic acid compositions are shown in Table 3. A total of three sugars and six organic acids were identified in GH3 at the ripening time. As can be seen from the proportion value, glucose content was the highest, followed by fructose, and the lowest was sucrose. Additionally, the levels of glucose and fructose were similar in the variety −49.73% and 48.10%, respectively (Table 3). The Kegg

(Kyoto Encyclopedia of Genes and Genomes) map of the sugar metabolism was mainly concentrated in glycolysis/gluconeogenesis (00010), starch and sucrose metabolism (00500), fructose and mannose metabolism (00051), galactose metabolism (00052), and amino sugar and nucleotide sugar metabolism (00520). Moreover, of all of the organic acids detected, citric acid had the highest proportion (49.40%), followed by succinic acid (49.40%), malic acid (29.43%), tartaric acid (12.66%), fumaric acid (1.29%), and the lowest was lactic acid (0.04%). The metabolic pathways of the organic acids in Kegg were mainly citrate cycle (Tricarboxylic acid cycle, TCA cycle; 00200), phenylalanine metabolism (00360), pyruvate metabolism (00620), glyoxylate and dicarboxylate metabolism (00630), carbon metabolism (01200), 2-oxocarboxylic acid metabolism (01210), and biosynthesis of amino acids (01230).

**Table 3.** The component characteristics of the sugars and organic acids of GH3 berries in the harvest stage.

| Compounds | | Proportion (%) | Kegg Map |
|---|---|---|---|
| Sugars | Glucose | 49.73 ± 0.74 | KO00010, KO00500, KO00052, KO00520 |
| | Fructose | 48.10 ± 0.14 | KO00051 |
| | Sucrose | 2.16 ± 0.02 | KO00500, KO00052, KO00520 |
| Organic Acids | Citric acid | 49.40 ± 4.41 | KO00020, KO00630, KO01200, KO01210, KO01230 |
| | Succinic acid | 29.43 ± 7.41 | KO00020, KO00360, KO00630, KO01200 |
| | Malic acid | 12.66 ± 2.41 | KO00020, KO00630, KO01200 |
| | Tartaric acid | 7.18 ± 4.02 | KO00630 |
| | Fumaric acid | 1.29 ± 0.26 | KO00020, KO01200 |
| | Lactic acid | 0.04 ± 0.01 | KO00620 |

The flavonoid and non-flavonoid compounds were analyzed to explore the characteristics of *V. adenoclada* GH3 from the composition (Table 4) and metabolic pathways (Figure 5). A total of 72 phenolic compounds were identified from the GH3 berry samples, each with three biological replicates: 12 benzoic acids, 2 benzeneacetic acids, 8 cinnamic acids, 1 resveratrol, 1 piceid, 26 flavonols, 6 flavan-3-ols, and 16 anthocyanins (Table 4). Flavonoids accounted for the highest proportion of the total phenols (84.06%), followed by stilbenes (13.55%), and the lowest was phenolic acids (2.40%). As regards the family of phenolic acids, it was observed that the main types were benzoic and cinnamic acids, and the proportion of benzeneacetic acids was only 4.50%. The Kegg map of phenolic acids was mainly concentrated in phenylpropanoid biosynthesis (00940). Only two types of stilbenes were quantified in GH3, namely, resveratrol and piceid, the proportions of which were 10.13% and 89.87%, respectively. The metabolic pathways of the organic acids in Kegg were stilbenoids, diarylheptanoids, and gingerol biosynthesis (00945). Of the three flavonoid substances, flavanols accounted for the highest proportion (73.26%), followed by anthocyanins (21.77%), and the lowest was flavan-3-ols (4.98%). The flavonoid-related Kegg pathways were mainly flavonoid biosynthesis (00941) and flavone and flavonol biosynthesis (00944).

**Table 4.** The component characteristics of the phenolic compounds of GH3 berries in the harvest stage.

| Phenolic Compounds | | Number | Proportion (%) | Kegg Map | Proportion of Total Phenolics (%) |
|---|---|---|---|---|---|
| Phenolic acids | Benzoic acids | 12 | 51.08 ± 2.44 | KO00940 | |
| | Benzeneacetic acids | 2 | 4.50 ± 0.23 | KO00940 | 2.40 ± 0.03 |
| | Cinnamic acids | 8 | 44.42 ± 2.54 | KO00940 | |
| Stilbenes | Resveratrol | 1 | 10.13 ± 0.83 | KO00945 | |
| | Piceid | 1 | 89.87 ± 0.83 | KO00945 | 13.55 ± 1.87 |
| Flavonoids | Flavonols | 26 | 73.26 ± 2.42 | KO00941, KO00944 | |
| | Flavan-3-ols | 6 | 4.98 ± 0.89 | KO00941 | 84.06 ± 1.90 |
| | Anthocyanins | 16 | 21.77 ± 2.99 | KO00941 | |

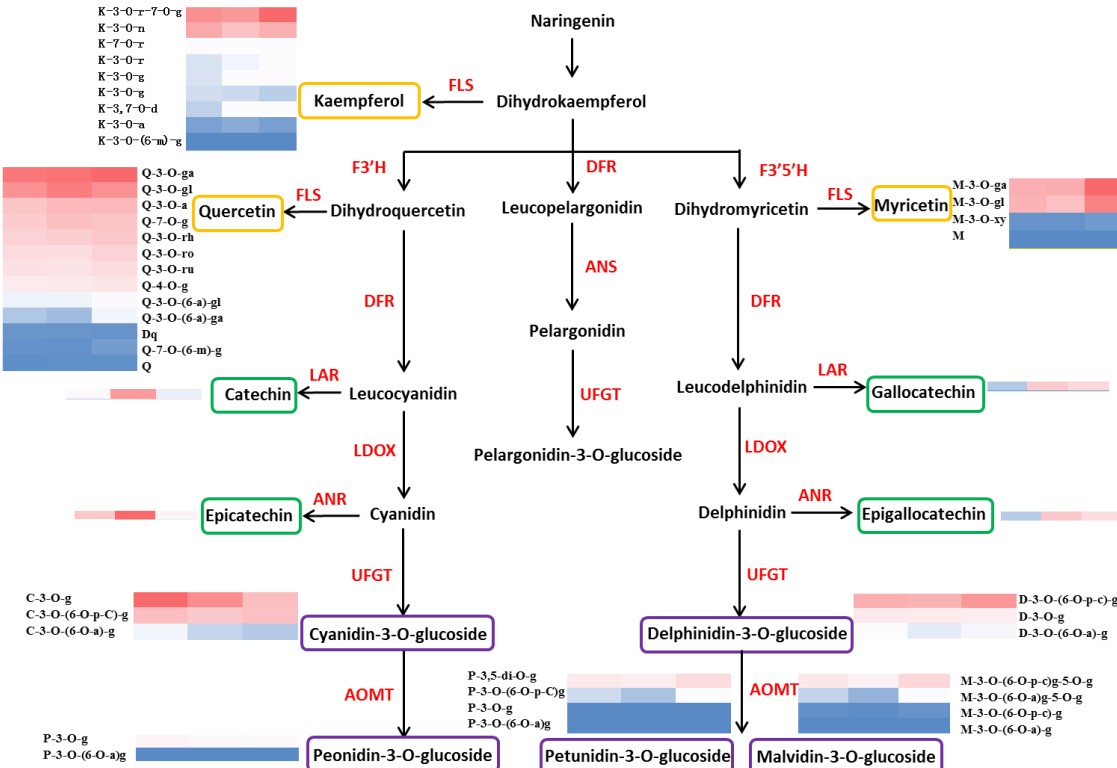

**Figure 5.** The metabolites in the flavonoid biosynthetic pathways in cv. GH3 in the mature stage. The grids with a color scale from blue to red represent content from low to high. F3'H, flavonoid 3'-hydroxylase; F3'5'H, flavonoid 3'5'-hydroxylase; FLS, flavonol synthesis; DFR, dihydroflavonol 4-reductase; LDOX, leucoanthocyanidin dioxygenase; LAR, leucocyanidin reductase; ANR, anthocyanin reductase; UFGT, UDP glucose-flavonoid 3-*O*-glcosyl-transferase; AOMT, anthocyanin *O*-methyltransferase. The three vertical columns of the heat map indicate three biological replications. Abbreviation for the phenolic compounds were illustrated in Supplement Table S2.

Figure 5 shows a simplified diagram of the flavonoid biosynthetic pathways and the comparison results of flavonols, flavanols, and anthocyanins in different types. In the flavonoid biosynthesis pathway, dihydrokaempferol, diquercetin, and dimyricetin were catalyzed to synthesize the flavonols kaempferol, quercetin, and myricetin by flavonol synthesis (FLS). Our results showed that 9 kaempferols, 13 quercetins, and 4 myricetins were detected in GH3 berries in the harvest stage. In addition, quercetins accounted for the highest proportion of total flavonols, followed by kaempferols, while the lowest was myricetins. Leucocyanidin reductase (LAR) and anthocyanin reductase (ANR) catalyze the production of catechin, gallocatechin, epicatechin, and epigallocatechin. Based on our results, epicatechin was the most abundant in all the flavan-3-ols, followed by catechin, and gallocatechin and epigallocatechin were both low in content. Of all the 16 anthocyanins we detected, there were three cyanidins, two peonidins, three delphinidins, four petunidins, and four malvidins, respectively. The content of cyanidin-3-*O*-glucoside was the highest, followed by delphinidin-3-*O*-(6-*O*-*p*-coumaroyl)-glucoside. F3'H (flavonoid 3'-hydroxylase) was involved in the biosynthetic pathway of quercetin derivatives, catechin, epicatechin, and cyanidin-based anthocyanins, and F3'5'H (flavonoid 3'5'-hydroxylase) was responsible for the synthesis of myricetin derivatives, gallocatechin, epigallocatechin, and delphinidin-based anthocyanins. As can be observed in our results, the content of dihydroxylated flavonoids catalyzed by F3'H was higher than that of trihydroxylated flavonoids.

## 4. Discussion

More than 70 species of grapevines have been reported across the world, which are distributed in three centers: The Eurasia–West Asia distribution center (*V. vinifera* L.), the North America distribution center (*V. labrusca* L., *V. riparia* Michx., and *V. rupestris* Scheele) and the East Asia distribution center (*V. amurensis* Rupr., *V. davidii* Foëx., *V. heyneana* Roem. & Schult, and other wild species) [11]. Located in the distribution center of East Asia, China is the center of origin of the richest grape genetic resources globally, with more than 40 known species of grapevine [12]. Chinese wild species have many excellent characteristics over *V. vinifera*, such as a high yield [12], disease resistance [12,13], a dark skin color [14], a rich aroma [11], and a good flavor [13].

*V. davidii* Foëx, *V. amurensis* Rupr, and *V. heyneana* Roem. & Schult are three of the most studied grape species in production and breeding. *V. adenoclada* Hand.–Mazz is a woody vine with similar morphological characteristics to *V. heyneana* Roem. & Schult, making the upper branches and leaves very difficult to distinguish [3]. However, according to systematic clustering analysis, *V. adenoclada* is categorized into the *V. heyneana* Roem. & Schult group. It should be emphasized that the main characteristic of *V. adenoclada* is the erect red–purple glandular hairs that grow on the tender shoots, and the mature glandular hairs are dark purple in color and hard in texture. Mature annual branches are still distributed at the base and are easily recognizable.

In the wild, *V. adenoclada* grapevines are generally distributed in shady environments, and they are vigorous in growth, resulting in a large annual growth. The surface of GH3 leaves is thick and smooth, which is similar to wild *V. adenoclada*, but both are smaller than *V. davidii* Foëx [8]. In addition, the leaves of wild *V. adenoclada* are 11–16 cm long and 6.5–12 cm wide, the new shoots are covered with gray arachnoid tomentum, and the base is covered in densely purple–brown glandular hairs, and the tendrils are branched and grow intermittently. All of this indicates that GH3 has typically botanical characteristics of the variety.

The quality of *V. adenoclada* is significantly different from that of other wild species, and each variety or superior line has its own characteristics. Although all three species have cylindrical or conical clusters, *V. adenoclada* grape generally has smaller and lighter clusters when compared with *V. amurensis* Rupr. and *V. davidii* Foëx [2,12,15]. The young fruit of GH3 is usually yellowish–green, and the skin finally appears purplish–black with the accumulation of anthocyanins after veraison. The above results are similar to the other *V. adenoclada* varieties. Jiang et al. [16] carried out a quality analysis on 10 species of wild grape from East Asia, including *V. adenoclada*. The results showed that the ripened berry and skin fresh weight of *V. davidii* Foëx were 2.60 g and 0.271 g, respectively, which are significantly higher than other Chinese wild grape species. The former researcher also indicated that a new type of *V. adenoclada* berries was round, and the average fresh weight was 1.34 g [8]. In the present study, the GH3 berry fresh weight reached 2.33 g at harvest time. Therefore, compared to other wild grapes, it might present more value in table grape breeding [17]. The results of previous studies showed that most of the wild grapes in China have the characteristics of low sugar and high acid, a TSS content less than 20, but a titratable acid content more than 10 [2,16]. The TSS content of a new type of *V. adenoclada* selected and bred by the research team of Hunan Agricultural University was up to 21 [8], which is close to the GH3 selected by our research team. Shuiyuan No.1 (GuiShenGuo No. 2012004) and Shuiyuan No. 11 (GuiShenGuo No. 2012003) obtained a variety certification organized by the Guangxi Crop Variety Certification Committee in 2012, which may be the female parents of GH3. The berry fresh weight of Shuiyuan No.1 and Shuiyuan No. 11 was about 1.81 and 2.20 g, and the TSS content of them was 15.3 and 11.5. Therefore, some characteristics relating to the quality of the new genotype GH3 were superior to that of its possible parents, cultivated in southern China.

The soluble sugar in grape berries is mainly composed of glucose and fructose, and some table grape varieties also have sucrose [18]. The results of previous studies on the sugar and acid components of Chinese wild grapes showed that the main sugars were glucose and fructose, while no sucrose was detected, and the fructose content was higher than glucose [16]. The results of this study showed that glucose and fructose were the main sugars in GH3, but a small amount of sucrose was also detected.

Organic acids play an important role in the flavor of grape berries, but variety, maturity, and climatic conditions greatly affect the content and distribution of organic acids [19,20]. Previous studies have shown that the main organic acids of Chinese wild grapes are tartaric acid and malic acid, accounting for approximately 90% of the total acids. Citric acid is below 10% of the total acids, and the content of that in *V. adenoclada* is the highest among all of the wild species [16]. However, in our research, citric acid showed the highest content among all of the organic acids, followed by succinic acid, malic acid, tartaric acid, fumaric acid, and the lowest was lactic acid in berries from harvest. Although the citric acid content in the grape juices produced from hybrid varieties is approximately 10 times lower than the content of the tartaric and malic acids [19], the former can inhibit the growth of yeasts in beverages [21]. Thus, GH3, as a new grape germplasm of *V. adenoclada,* has remarkable characteristics in the composition of sugars and acids.

Each grape variety has its own phenolic composition characteristics, and this is largely determined by genotype [16,22]. However, the content of phenolic compounds is influenced by a series of external factors, such as the climatic conditions, the degree of ripeness, the training systems, and the cultural practices [16,23–25]. Previous studies on the composition of phenolic substances of new hybrid varieties grown on two rootstocks showed that the content of phenolic acids corresponded to 2.33–14.3% of the total phenolic compounds [19]. In our research, the proportion of phenolic acids in GH3 berries was 2.40%, which is lower than that of stilbenes and flavonoids. However, phenolic acids were found to be the most abundant polyphenolic compound in the Kyoho grape [26]. Generally, resveratrol and its glycosides were abundant in the skin of grapes, while a substantial proportion of resveratrol was in the form of its glycoside, called piceid. It is generally known that resveratrol has notable potential health benefits and has more bioavailability than piceid [27]. The current study suggested that the ratio of piceid to resveratrol was approximately 9:1 in GH3 berries.

Flavonoids, as antioxidants, include several classes mainly found in the seeds and skin of grapes, such as flavonols, flavanols, and anthocyanins [24]. In our study, quercetin, kaempferol, and myricetin were the most detected flavonols. The research of Jiang Yu et al. [16] showed that among different species of Chinese wild grape, there are big differences in their anthocyanin profiles, which is more related to the expression of the F3′5′H, F3′H, and the AOMT genes [28]. As can be seen from our research results, cyanidin-3-*O*-glucoside presented the highest amount in the GH3 berries, thought to be the characteristic anthocyanin of the variety. Moreover, the metabolic pathway related to B-ring dihydroxylation was more active than trihydroxylation in GH3 berries. For this reason, the differences in the flavonoid profile are more related to the gene expression in the synthesis pathway.

## 5. Conclusions

The breeding process, botanical characteristics, cultivation and management techniques, and quality-related parameters of GH3 were investigated in this study to evaluate its characteristics as a new *V. adenoclada*. This is the first study to use widely targeted metabolomic analysis methods to study the quality-related indicators of *V. adenoclada*. The results indicated that GH3 has the typical botanical characteristics of *V. adenoclada*, but its berries are bigger, and its sugar content is higher than those of wild or other *V. adenoclada* varieties. On the other hand, the main sugar and acid components of GH3 grape were shown to be glucose and citric acid, respectively. Moreover, cyanidin-3-*O*-glucoside presented as the characteristic anthocyanin of the variety. In addition, the B-ring dihydroxylation demonstrated more activity than trihydroxylation from the metabolic pathway of flavonoids. Therefore, it can be seen from the present study the genetic background of GH3 has its own characteristics, which will be further explored in our follow-up study.

**Supplementary Materials:** The following are available online at http://www.mdpi.com/2073-4395/10/12/1851/s1, Table S1: Some physical and chemical parameters of GH3 berries at harvest stages in 2018 and 2019, Table S2: Abbreviations for the phenolic compounds used in Figure 5.

**Author Contributions:** Conceptualization, D.-D.W. and G.C.; methodology, G.C., D.-D.W., and S.-H.Z.; investigation, H.-Y.L., N.Y., and J.Z.; resources, D.-D.W.; data curation, G.C., S.-H.Z., and H.-Y.L.; writing—original draft preparation, D.-D.W. and G.C.; writing—review and editing, G.C., S.-H.Z., and L.-J.X.; supervision, D.-D.W.; project administration, D.-D.W. and G.C.; funding acquisition, D.-D.W. and G.C. All authors have read and agreed to the published version of the manuscript.

**Funding:** This research was funded by the National Nature Science Fund Project (grant number 31860529), the integrated demonstration of key techniques for the industrial development of featured crops in rocky desertification areas of Yunnan–Guangxi–Guizhou provinces (grant number SMH2019-2021), the Guangxi Key Research & Development Plan (grant number GuikeAB18294032), and the Guangxi Luocheng County Experimental Station of *V. quinquangularis* Rehd. grape (grant number GuiTS201418).

**Acknowledgments:** The authors want to acknowledge the support in the investigation of botanical characteristics by Lan Xiangye, who works in the Agriculture Bureau of Guangxi Luocheng County.

**Conflicts of Interest:** The authors declare no conflict of interest.

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
