# Peer review of "The Cultivation Techniques and Quality Characteristics of a New Germplasm of Vitis adenoclada Hand.-Mazz Grape"

_agronomy, doi:10.3390/agronomy10121851_

Round 1

Reviewer 1 Report

   An interesting paper. While the potential risks to the organism posed by the genetically modified crop, the grape GH3 in this paper, should be carefully considered by further research, the metabolome analysis of that is one of the important studies for the chemical quality assessment. In addition, the botanical characteristics of GH3 were discussed with those of the wild species.

(1) Fig. 5: What do the three vertical columns of the heat map indicate?

(2) According to The Plant List (http://www.theplantlist.org/), the name Vitis adenoclada Hand.-Mazz may not have been approved. In addition, V. quinquangularis may be a synonym of V. heyneana Roem. & Schult. Check them again.

Reviewer 2 Report

The manuscript of Wu characterizes Vitis adenoclada Hand.-Mazz giving plenty of details from the botanical point of view and included tips for plant cultivation and management. Next, the authors analysed berry primary and secondary metabolites.

I think the manuscript gives new information regarding this Asian species and it is worth publication. Below some commentaries and suggestions.

Line 20. I suggest changing the word metabolome with metabolites since we are talking of one targeted analysis. Same when it is mentioned in the conclusion part.

Line 49-53. Please explain better what this sentence means. It is not clear to me, what does means the seeds were “mixed”. Did you collect the pollen from the five vines and manually pollinated the Shuiyuan wild grape?

Line 57. Which are the dual purposes?

Line 58-605. Literature studies to cite? Where is this information coming from? The sentences are part of the introduction and I think references are necessary. if this knowledge is coming from the study under evaluation, I suggest moving it to in the results section accompanied by analytical data.

Line 68. Identification

Line 89. The words “were adopted” does not grammatically agree with the sentence.

Line 111. The freeze-dry sample is the remaining 50 berries for the first 4 samplings? And what about the fifth, since you used 300 for the physiological measurements? These berries were with or without seeds? Please explain better this part.

Paragraph 3.4 it is important to specify and underline that the metabolites analyses were done at the ripe stage, that the quantity of sugar and acids.

Line 353. The weight is referred to as the ripe berry? Please specify.

Line 372-374. Also here please specify the developmental stage.

Reviewer 3 Report

In this paper, that is well written and easily readable, authors describe the main botanical, physical, chemical and metabolomic characteristics of the new genotype GH3, a natural hybrid of the wild V. adenoclada grapevine. The topic covered in the paper is in line with one the main objective of Sustainable Viticulture, as is the introduction of new varieties, characterized by both good quantitative and qualitative characteristics and source of resistance against abiotic and biotic adversities. However, in my opinion, the paper has different disadvantage that makes it not publishable in a scientific journal as Agronomy. In particular:

  1. Authors detected parameters, such as berry fresh weight, total soluble solids (TSS) content, pH, titratable acidity, color indices (paragraph 2.3) and metabolomic components of berries (paragraph 2.4), during a single year of experimentation. Since the expression these parameters, unlike the botanical ones, is strongly influenced by the environmental component, their appropriate evaluation requires repeated experimentation over time (minimum two years).
  2. As stated in the previous point, the chemical and metabolomic parameters considered by authors are strongly influenced by the environmental and cultivation conditions. Therefore, in my opinion, a more useful approach would have been to compare this different parameters detected both on the genotype GH3 and its parent of adenoclada in the same site of cultivation. In this way, the authors could have better discuss any differences and highlight the improving aspects of the new genotype compared to the parental.

Major revisions:

  1. Introduction – lines 57-65: authors report that the new grapevine genotype GH3 is particularly adaptable to the climatic conditions of southern China. My question is: where do these considerations came from? From a previous experimental activity? If yes, I suggest to the authors to report the bibliographic references, in the same way as for adenoclada at lines 66-69.
  2. Material and method: the authors do not describe the statistical approach used for the analysis of the results.
  3. Results - Paragraph 3.1.3: authors report as results some characteristics of the new grapevine genotype GH3, related to their resistance against biotic and abiotic adversities. However in the section “Materials and Methods” authors did not describe the methodology used to evaluate these resistance/tolerance characters. For example, the phenotypic evaluation of resistance to biotic disease (such as anthracnose, grey mold, powdery mildew, etc.) can be performed by scoring the presence or absence of visible symptoms in the phenological phases in which the plant is more susceptible to infection and by using specific index in order to evaluate the severity of disease. What emerges from reading the paper is that these phenotypic characters have been detected through simple observation, which does not represent a correct scientific approach.
  4. Results – Paragraph 3.3: authors describe some techniques for the cultivation and management of the new grapevine genotype GH3, without reporting any results deriving from experimental activity. In my opinion this paragraph should be integrated in the Introduction (for example between lines 57 and 65), by reporting specific bibliographic references.

Minor revisions

  1. The name of a species must be reported in its complete form (e.g. Vitis adenoclada Hand-Mazz.) only the first time; subsequent times it must be reported in its abbreviated form (e.g. adenoclada) – a general check throughout the document is recommended
  2. Line 58: remove the sentence “is highly adaptable”
  3. Line 94: probably is “Brix start increasing”
  4. Line 101: replace the sentence “in the harvest stage” with “at the harvest stage”
  5. Check the bibliographic reference n. 6, probably the title is not correct.

Round 2

Reviewer 3 Report

Response to authors Comments

In this paper, that is well written and easily readable, authors describe the main botanical, physical, chemical and metabolomic characteristics of the new genotype GH3, a natural hybrid of the wild V. adenoclada grapevine. The topic covered in the paper is in line with one the main objective of Sustainable Viticulture, as is the introduction of new varieties, characterized by both good quantitative and qualitative characteristics and source of resistance against abiotic and biotic adversities. However, in my opinion, the paper has different disadvantage that makes it not publishable in a scientific journal as Agronomy. In particular:

Point 1: Authors detected parameters, such as berry fresh weight, total soluble solids (TSS) content, pH, titratable acidity, color indices (paragraph 2.3) and metabolomic components of berries (paragraph 2.4), during a single year of experimentation. Since the expression these parameters, unlike the botanical ones, is strongly influenced by the environmental component, their appropriate evaluation requires repeated experimentation over time (minimum two years).

Response 1:Physical and chemical indexes are indeed greatly affected by environmental factors. Some research in the field of cultivation has been repeated for many years(minimum two years). GH3 as a new germplasmof V. adenoclada Hand.-Mazz grape, we have also observed fruit quality indicators for many years. But in previous years, all the experimentation were focused on the fruits from the harvest stage, there is no systematic monitoring of several developmental stages. In the present research, we collected grape berries in three biological replicates in five E-L stages. Thus, berry fresh weight, total soluble solids (TSS) content, pH, titratable acidity, color indices could be detected at differentdevelopmental stages. In addition, the focus of this study is to demonstrate the characteristics of GH3 without other field treatments or comparison of multiple varieties. So we used a single year of environmental data.

Response 1: If the authors have results regarding the physical, chemical and metabolomic data regarding the new GH3 genotype (even if focused at the harvest stage) , obtained in years prior to the one described in the paper, I suggest to include them. For example, authors could describe the activities carried out in previous years in a new paragraph of the section "materials and methods" and report the table of results in the Supplementary materials. In this way, a reader is able to understand how the selection of the new grapevine genotype GH3 was based on observations repeated over time that confirmed its goodness. Finally, I have not fully understood the last two sentences made by the authors: " In addition, the focus of this study is to demonstrate the characteristics of GH3 without other field treatments or comparison of multiple varieties. So we used a single year of environmental data". Even if the goal of a experimentation is to demonstrate the characteristics of a new genotype (with or without treatments, etc.) by using physical and chemical index, that are strongly influenced by environment, several years of observation are useful.

Point 2: As stated in the previous point, the chemical and metabolomic parameters considered by authors are strongly influenced by the environmental and cultivation conditions. Therefore, in my opinion, a more useful approach would have been to compare this different parameters detected both on the genotype GH3 and its parent of adenoclada in the same site of cultivation. In this way, the authors could have better discuss any differences and highlight the improving aspects of the new genotype compared to the parental.

Response 2:As the reviewer suggested, we could better elucidate the advantages of the variety as a new genotype compared to the parental. However, as we mentioned in the introduction, we do not know accuratefemaleparent of GH3.In 2011, the natural hybrid seeds of five lines of pistillate flower of V. adenoclada Hand.-Mazz grape, such as Shuiyuan No. 1, Shuiyuan No. 11, Jiulong No. 1, Cotton No. 5, and Lile No. 6, were mixed and seeded together in the Shuiyuan wild grape base in Luochengcounty, and 139 seedlings were planted in an open field. Thus, the father of natural pollination is unknown.

Response 2: in their manuscript the authors introduce the GH3 as a new grapevine genotype of V. adenoclada, with different characteristics that are better than the known parent. Therefore, in my opinion the authors in the Discussion section should have compared the main characteristics of the new genotype GH3 with those of the only know parent and/or other wild grapevine species, cultivated in the same environmental (southern Cina). 

Point 3: Introduction – lines 57-65: authors report that the new grapevine genotype GH3 is particularly adaptable to the climatic conditions of southern China. My question is: where do these considerations came from? From a previous experimental activity? If yes, I suggest to the authors to report the bibliographic references, in the same way as for adenoclada at lines 66-69.

Response 3:GH3 belongs to V. adenoclada Hand.-Mazz grape, so it has some characteristics of the species, such as Adapt to the southern climate conditions, suitable for cultivation in karst mountainous areas, etc. As the reviewer suggested, we have added some relevantreferences.

Response 3: I have not further comments

Point 4:Material and method: the authors do not describe the statistical approach used for the analysis of the results.

Response 4:Because there was no ‘statistical approach’part in agronomy-template. But we have added ‘Statistical analysis’ as 2.5 in Materials and methods now in revised manuscript.

Response 4: I suggest to eliminate the last sentences, line 149

Point 5:Results - Paragraph 3.1.3: authors report as results some characteristics of the new grapevine genotype GH3, related to their resistance against biotic and abiotic adversities. However in the section “Materials and Methods” authors did not describe the methodology used to evaluate these resistance/tolerance characters. For example, the phenotypic evaluation of resistance to biotic disease (such as anthracnose, grey mold, powdery mildew, etc.) can be performed by scoring the presence or absence of visible symptoms in the phenological phases in which the plant is more susceptible to infection and by using specific index in order to evaluate the severity of disease. What emerges from reading the paper is that these phenotypic characters have been detected through simple observation, which does not represent a correct scientific approach.

Response 5:We admit that this part of the content is not supported by data, just through field observation. Results - Paragraph 3.1.3: Stress ResistanceThere should be corresponding data support and method introduction as the research results. We have the following explanations: 1. Investigation of disease resistance is not the focus of the present study. We just introduced briefly the characteristics of GH3. 2. GH3 as a new germplasmof V. adenoclada Hand.-Mazz grape, there is a lot of research that has not been done yet. We are in the process of conducting detailed disease resistance tests. If the reviewer and editor agree to delete this part of the content, we also agree.

Response 5: because the authors are in the process of conducting detailed disease resistance test, I suggest to this part of the content

Point 6:Results – Paragraph 3.3: authors describe some techniques for the cultivation and management of the new grapevine genotype GH3, without reporting any results deriving from experimental activity. In my opinion this paragraph should be integrated in the Introduction (for example between lines 57 and 65), by reporting specific bibliographic references.

Response 6:Results – Paragraph 3.2: Although we do not reportany results deriving from experimental activity of techniques for the cultivation and management, this was indeed part of the study, especially since this is the first time the varietymight been publicly reported. We ask the reviewers to consider allowing us to retain Paragraph 3.2 in results

Response 6: In my opinion, a simple description of the cultivation techniques, that can be adopted for the management of a specific variety of grapevine, cannot be considered as a result of a experimental work. If the different techniques described in the paragraph 3.2 have been used by the authors for the management of their experimental field, they can insert this paragraph in the section "Material and Methods"

Point 7: The name of a species must be reported in its complete form (e.g. Vitisadenoclada Hand-Mazz.) only the first time; subsequent times it must be reported in its abbreviated form (e.g.adenoclada) – a general check throughout the document is recommended

Response 7:We have already modified them in modified manuscript.

Response 7: I'm sorry; there was an error when I copied my comments at the first step of the review. So I rewrite the comment n. 7 in their correct form: "The name of a species must be reported in its complete form (e.g. Vitis adenoclada Hand-Mazz.) only the first time; subsequent times it must be reported in its abbreviated form (e.g. V. adenoclada) – a general check throughout the document is recommended"- this comment is also valid for the scientific names of the other varieties reported in the paper and not only for the V. adenoclada.

Point 8:Line 58: remove the sentence “is highly adaptable”

Response 8:We have already removed the sentence “is highly adaptable”. Line 58-59

Response 8: I have not further comments

Point 9:Line 94: probably is “Brix start increasing”

Response 9:We have already modified them in modified manuscript. Line 96

Response 9: I have not further comments

Point 10:Line 101: replace the sentence “in the harvest stage” with “at the harvest stage”

Response 10:We have already modified them in modified manuscript. Line 105

Response 10: I have not further comments

Point 11:Check the bibliographic reference n. 6, probably the title is not correct.

Response 11:We have checked the bibliographic reference n. 6, and not clear which reference the reviewer refers to.

Response 11: I refer to the following reference: "Li, H.X.; Chen, H.; Wang, Z.; Zhong, X.H.; Wang, X.R.; Liu, K.Y. Observation and study on agronomic characters of new type of Vitis adenoclada Hand.–Mazz. Northern Horticulture 2018, 12, 63-67." - on the website of Northern Horticulture is reported the following title: "
A Survey Study on the New Types of Agronomic Characteristic of Vitis adenoclona Hand.-Mazz"
